# A Systematic Review Protocol of Opportunities for Noncommunicable Disease Prevention via Public Space Initiatives in African Cities

**DOI:** 10.3390/ijerph19042285

**Published:** 2022-02-17

**Authors:** Ebele R. I. Mogo, Taibat Lawanson, Louise Foley, Clarisse Mapa-Tassou, Felix Assah, Toluwalope Ogunro, Victor Onifade, Damilola Odekunle, Richard Unuigboje, Nfondoh Blanche, Rose Alani, Lia Chatzidiakou, Olalekan Popoola, Roderic Jones, Tolu Oni

**Affiliations:** 1Global Diet and Activity Research Group and Network, MRC Epidemiology Unit, University of Cambridge, Cambridge CB2 0SL, UK; ebele.mogo@mrc-epid.cam.ac.uk (E.R.I.M.); louise.foley@mrc-epid.cam.ac.uk (L.F.); tolu.oni@mrc-epid.cam.ac.uk (T.O.); 2Department of Regional and Urban Planning, University of Lagos, Lagos 101017, Nigeria; tlawanson@unilag.edu.ng (T.L.); vonifade@unilag.edu.ng (V.O.); dodekunle@unilag.edu.ng (D.O.); richardunuigboje@gmail.com (R.U.); 3Center for Housing and Sustainable Development, University of Lagos, Lagos 101017, Nigeria; rehobotty@gmail.com; 4Health of Populations in Transition Research Group (HoPiT), University of Yaoundé I, Yaounde 8046, Cameroon; mapatassou@gmail.com (C.M.-T.); nfondohblanche@gmail.com (N.B.); 5Air Quality Monitoring Research Group, Department of Chemistry, University of Lagos, Lagos 101017, Nigeria; ralani@unilag.edu.ng; 6Yusuf Hamied Department of Chemistry, University of Cambridge, Cambridge CB2 1EW, UK; ec571@cam.ac.uk (L.C.); oamp2@cam.ac.uk (O.P.); rlj1001@cam.ac.uk (R.J.)

**Keywords:** public spaces, healthy cities, urban health, healthy environments, built environment, African cities, health promotion

## Abstract

Public spaces have the potential to produce equitable improvements in population health. This mixed-methods systematic review aims to understand the components of, determinants, risks, and outcomes associated with public space initiatives in African cities. This study will include quantitative and qualitative study designs that describe public space initiatives in African cities with implications for promoting health and wellbeing, particularly through the prevention of noncommunicable diseases. Only studies published after 1990 and that contain primary or secondary data will be included in the review. Literature search strategies will be developed with a medical librarian. We will search PubMed, using both text words and medical subject headings. We will adapt this search to Scopus, Global Health, and Web of Science. This systematic review will adopt a mixed methods analytical approach. Mixing will occur in extracting both qualitative and quantitative findings; in synthesizing findings; and in the analysis where we will integrate the qualitative and quantitative strands. The learnings from this study will contribute to advancing knowledge on noncommunicable disease prevention through public space initiatives in African cities.

## 1. Introduction

In the African region, comprehensive approaches are crucial to ensure that good health becomes an outcome of the fast-paced urbanisation occurring across the continent. The United Nations’ Statistical Commission’s international method of comparison for cities defines cities as settlements with a population of at least 50,000 dwellers who live in contiguous dense grid cells that have over 1500 inhabitants per square kilometre [1]. The World Health Organization defines healthy city initiatives as efforts that create and improve (I) physical environments, (II) social environments, and (III) community resources, with the end goal of ensuring the optimal development of urban residents [2].

Current statistics place Africa’s urban growth rate as one of the fastest in the world. Urbanisation on the continent is consistently typified by high poverty, high socio-economic inequalities, and insufficient resources, basic services, and FF infrastructure necessary for a decent life. These factors worsen the double-burden of infectious and noncommunicable diseases and lead to a cumulative urban disadvantage that contributes to reduced quality of life and wellbeing in African cities [3,4].

In light of the limited funds to address the health and social consequences of unplanned and inequitable urbanisation, it is important to consider high-impact, city-scale interventions that can rapidly alter the status quo, to produce health and wellbeing gains. Public spaces are typified by being open, accessible spaces that lie outside the sole control of individuals and in which people engage in individual or group activities [5]. They may include but not be limited to built spaces, such as parks and stadiums, natural spaces, such as hills and coasts, blue spaces, such as beaches and public swimming pools, and informal spaces, such as junctions and spaces under bridges that the public may appropriate for initiatives.

Public spaces have the potential to cut across diverse socioeconomic and spatial segments of urban residents to produce equitable improvements in population health. Examples of such spaces include streets, bridges, roundabouts, sidewalks, blue spaces such as beaches, and green spaces such as parks. They can serve as a facilitator of health-promoting resources, services, and behaviours [6], while regulating and halting the negative externalities of urbanisation such as pollution via green spaces [7], injury through well designed public spaces, and mental illness and obesity via recreational spaces that promote physical activity and community participation.

The design of, access to, and condition of urban infrastructure in public spaces is a critical influence on the social and behavioural factors that shape quality of life, health, and wellbeing in cities. When properly designed and equitably spread, they can promote health and wellbeing in a variety of ways. Infrastructural provisions such as parks, green spaces, sidewalks, bike lanes, amongst others, can ensure that people can engage in safe physical activity [8] and in social activities that promote mental health, facilitate access to needed resources, and ensure that people can engage safely in utilitarian and recreational activities [9].

On the other hand, where inadequately designed and/or poorly maintained, infrastructure can pose multipronged risks to health, wellbeing, and quality of life. For example, in many cities across Africa, insufficient provision of high-quality infrastructure needed to engage in leisure physical activity in public spaces leads many residents to appropriate existing infrastructure such as roundabouts, spaces under bridges, and streets for physical activity with the likelihood for increased injury risks. Therefore, while physical activity is key to reducing the risk for several chronic diseases, residents could become exposed to other harmful factors such as air pollution, environmental waste, injury, and safety risks in the course of engaging in physical activity in spaces that are not conducive for these behaviours. As such, while residents may potentially gain health and social benefits from using public spaces, these benefits may be outweighed by unintended negative consequences to their physical and social wellbeing.

Currently, an overview of the nature of infrastructural provisions in urban public spaces in Africa and their impact on noncommunicable-disease-related health outcomes and behaviours such as leisure time physical activity, mental health, and healthy diets, is lacking. Such information is needed to understand how urban infrastructure can be optimised as a high-impact investment for health on the continent. This systematic review aims to understand the components, continental diversity of, and associated health outcomes of public space initiatives targeting the prevention of noncommunicable diseases in African cities. The knowledge yielded will contribute to advancing and informing urban health promotion, particularly chronic disease prevention, through public space infrastructure as well as strategies for designing safe and multifunctional public spaces.

Our specific aims are to:Synthesise peer-reviewed and grey literature on the design, components, outcomes and variety of public space initiatives in African cities to document initiatives with relevance for noncommunicable disease prevention;Inform the discourse on how public spaces in rapidly urbanising African communities can be optimised for health promotion.

In line with our specific aims, our research questions are thus:What is the existing evidence on the design, components, outcomes and variety of public space initiatives in African cities, with relevance to the prevention of noncommunicable diseases?How can public spaces in rapidly urbanising African communities be enhanced for the prevention of noncommunicable diseases, for example, through the promotion of mental health, physical activity, healthy diets, and the reduction of injuries?

## 2. Materials and Methods

This mixed-methods systematic review will be conducted in accordance with PRISMA guidelines [10] and has been registered with the International Prospective Register of Systematic Reviews (PROSPERO) under registration number CRD42020189285.

Our study is conceptually informed by the World Health Organization and UN-Habitat framework for integrating health into urban and territorial planning [11]. This framework approaches health integration into public spaces by considering the settings where initiatives are implemented, principles driving implementation, sectors involved, and outcomes targeted as entry points for health creation.

### 2.1. Eligibility Criteria

#### 2.1.1. Types of Studies

We will systematically synthesise existing primary quantitative and qualitative evidence in academic and grey literature on public space initiatives in Africa, with a focus on noncommunicable disease prevention in urban contexts. Studies to be selected must have been conducted in 55 member states of the African Union [12], published since 1990, and contain and analyse primary or secondary data. We have chosen this duration as it coincides with the period when the promotion of the concept of healthy cities began [13]. We will also include reports, manuals, and guidelines on the development and implementation of specific initiatives, program manuals and guidelines, and policy documents aligned with the research questions.

We will exclude studies that report initiatives that do not take place in an African city or that were implemented before 1990. We will also exclude literature reviews, commentaries, opinion pieces and narrative overviews that describe public space initiatives, but we will use these to provide context and to identify primary literature as well as cues for identifying grey literature. At a previous stakeholder engagement initiative, multisector decisionmakers from East, West, Central and South Africa pointed out the need for peer-reviewed research with a stronger emphasis on solutions and initiatives to address non-communicable disease (NCD) risk factors in urban settings [14]. An analysis of existing peer-reviewed studies will help us identify the current knowledge, gaps, and opportunities in the research. While primarily analysing peer-reviewed literature in this study, we will also use grey literature to provide insights as we transform our findings into recommendations for policy, action, and transdisciplinary research.

#### 2.1.2. Types of Interventions

Initiatives included in this study must focus on modifying public spaces in African cities for the prevention of noncommunicable diseases. These could include (1) the design, implementation and/or maintenance of physical components of public spaces, e.g., through green space development or (2) the design, implementation, and/or maintenance of social aspects of public spaces with implications for addressing NCD risk factors, such as the governance dimensions of community gardening, the organizational aspects of social recreational activities and group exercise activities, or related interventions. We have included not only the physical infrastructure but also social dimensions of the use of public space for NCD prevention. Both factors influence the impact and use of public spaces for health promoting behaviours and for wellbeing [15], and thus will provide a richer understanding of the conditions under which people use public spaces for health promotion.

#### 2.1.3. Types of Participants

We will not limit the age, gender, and ethnicity of the populations targeted by studies included in this systematic review. All study settings and initiatives run by any sectors, e.g., education, environment, health will be included as long as they take place in public spaces and relate to noncommunicable disease prevention in African cities. We have chosen to include literature from multiple sectors and settings given that decisions on the use, design, and maintenance of public spaces in African cities are made by actors from multiple disciplines [16].

#### 2.1.4. Types of Outcome Measures

We will report outcome measures for studies that meet the inclusion criteria listed above, including quantitative and qualitative outcomes. Primary outcome measures will include any objective measures of health (e.g., reduction in hypertension prevalence) or social outcomes (improved participation in recreational activities) associated with public space initiatives. Secondary outcome measures will include health behaviours, e.g., walking, and any measured health exposures, e.g., exposure to air pollution. We will also document how outcomes were assessed and enabling or limiting factors that could inform future replication. This ties into our goal of ensuring that the findings of our study will be able to meet the voiced needs of multisectoral stakeholders in urban Africa for understanding the “how” of such initiatives, particularly as they concern the design, scale, maintenance, and implementation of urban initiatives.

### 2.2. Search Strategy

Literature search strategies will be developed using medical subject headings and text words, as well as an extensive list of identified databases to be searched. We will enlist the help of a medical librarian in developing a search strategy in PubMed. We will adapt this search to SCOPUS, Global Health, and Web of Science. The search strategy for each of these databases is included in the Appendix A. We have chosen each of these databases considering the multidisciplinary and global nature of our inquiry and with the goal of covering literature in the medical, public health, and social science literature. We will include all languages, and use Google Translate as well as the expertise of the research team and collaborating institutions for translation. We expect that most articles will be published in English, French, Spanish, Portuguese, or Arabic, as these languages are the most frequently used for empirical documentation in African countries.

Our grey literature search will involve searching grey literature databases, internet searches of Google and targeted websites, and by consulting with key contacts in government and nongovernmental agencies, and academic topic experts. Our internet search will include identifying reports from local, regional, and international agencies such as the WHO, UN-Habitat, UNICEF, amongst others, that have a focus on public spaces and/or health in African cities. We will use Google (in incognito mode) for this search, and then narrow our search by country, domain, and file type. We will inform stakeholders across the key partner universities of our initial findings and then work with them to identify additional sources of literature, particularly grey literature, to further contextualise our findings and ensure our search is as exhaustive as possible. The emerging findings will also serve the additional goal of facilitating ongoing transdisciplinary inquiry and knowledge coproduction on the topic.

### 2.3. Study Screening and Selection

We will export our search results into EndNote and remove duplicates before uploading them into Covidence, a digital systematic review platform, for further review, extraction, and quality assessment. We will assemble a team of researchers across the project team to support the review. Teams will double-screen 100% of the titles and abstracts, and subsequently full-text articles, according to our inclusion criteria. Where conflicts arise, inclusion and exclusion criteria, including definitions of interventions/exposures and outcomes will be reviewed, clarified, and refined until conflicts are resolved. Where there is no resolution, a third coinvestigator will be brought in to clarify any outstanding conflicts and adjudicate.

### 2.4. Data Extraction

This will be followed by extraction for articles that meet the inclusion/exclusion criteria. A senior researcher will design a data extraction template and pilot it with another senior researcher, via doubly extracting 5% of the articles. This template has been added to the Appendix A. Their responses will be compared and used to modify the data extraction template where necessary. Junior researchers will then be assigned full text articles to extract, using the finalised data extraction template. A minimum of 10% of selected data fields in the final articles will be doubly extracted, concordance will be checked, and the proportion of double extraction increased if necessary. For non-English studies, group members with the necessary language competence will be asked to accept or reject articles using the inclusion or exclusion criteria. For studies where there is no group expertise on the language, we will make use of Google Translate. The reference list of identified articles will be used to identify more references for backward and forward reference search.

The following information will be noted: (I) inclusion and exclusion criteria for participation where applicable; (II) types of funding and implementation partners; (III) dates of project implementation, publication and data collection; (IV) institutional and country affiliations of authors; (V) sources, methodologies, and types of evidence used; (VI) the nature, type, setting, spatial level, and targeted outcomes of public space initiatives in public spaces in African cities; (VII) population demographic characteristics; (VIII) details around the long-term sustainability of the project as well as lessons, barriers, and facilitators. These extraction dimensions were informed through input from stakeholders intersecting health, planning, environment, and governance in African cities with the goal of addressing existing knowledge gaps on urban evidence generation for urban health.

### 2.5. Assessing Study Quality

Standardised templates will be used to extract data with every eligible paper reviewed by two researchers. Approximately five senior researchers will appraise all studies that qualify for extraction. We will make use of the qualitative checklist and the cohort study checklist of the Critical Appraisal Skills Programme (CASP) [17]. There are multiple appraisal tools in the literature [18]. However, we chose the Critical Appraisal Skills Programme (CASP) tool given that it is the most used appraisal template for appraising and synthesizing evidence on public health related topics, recommended by the Cochrane Qualitative and Implementation Methods Group as well as the World Health Organization [19]. Additionally, it is designed to be applied in pedagogic, workshop, and educational settings, and thus would be valuable for building the capacity of our research team, while also supporting the use of our research in shaping future transdisciplinary research and action in similar spaces.

The instrument will be modified to accommodate cross-sectional studies. The modified template will capture the following aspects of quantitative studies: (I) whether the study was focused in its approach; (II) the suitability of the recruitment of the cohort; (III) the extent to which bias was measured and minimised; (IV) the consideration given to confounding in the design and analysis of the study and the implications of the study for practice. It will also consider the following aspects of qualitative studies: (I) the appropriateness of a qualitative approach where used; (II) the appropriateness of the design, recruitment and data collection strategy; (III) the consideration given to positionality between the researcher and participants; (IV) the consideration given to ethical issues, the rigour, clarity, and value of the findings. We anticipate varying levels of robustness of the data at the quality appraisal stage due to the heterogeneity of disciplines and papers we will consider. While we do not envisage excluding studies or applying thresholds based on study quality, we aim to give more weight to information from more comprehensive studies in refining our interpretation. Most importantly, we aim to use the appraisal tool to identify and communicate quality issues that recur across the literature to inform the design of future initiatives. This approach mirrors that taken in a previous study the research team conducted on gender and socioeconomic dimensions of public space use for transport in urban Africa [20].

### 2.6. Analysis

Mixed-methods systematic reviews integrate qualitative and quantitative research findings and enhance the utility and impact of systematic reviews for influencing policy and practice [21]. They also help to better represent the experiences of the communities which the research aims to serve. This systematic review will adopt a mixed-methods analytical approach in three ways: in the extraction of both qualitative and quantitative findings, in the synthesis of findings, and in the process of analysis.

The analytic plan will be finalised after the screening process. We do not anticipate having sufficient information to conduct a quantitative meta-analysis due to the wide range of study designs, analytic units, and methods of assessment. However, we will be informed by the Bradford Hill criteria [22] in exploring the strength, consistency, and plausibility of the studies to infer the relationships that could be causal. In this process, we aim to use a parallel convergent method [23] to synthesise the parallel strands of our study. Compared to an exploratory sequential design where a subsequent quantitative phase is used to test tools whose development is informed by qualitative data, or an explanatory sequential design where the qualitative data are used to contextualise the findings of quantitative data, we have chosen a parallel convergent approach as it helps us to weigh the qualitative and quantitative findings concurrently, to compare them and allow them to simultaneously enrich one another [24]. We have chosen a thematic synthesis as our approach for synthesising the evidence gained due to its appropriateness for synthesising evidence with the aim of informing interventions [25,26]. This approach is particularly appropriate for integrating data with various categories, such as the information on the location, impact, evaluation, and partnerships in our case. The three steps of the thematic synthesis, namely the data coding, the development of descriptive themes, and the generation of analytic themes which will be transformed into policy and intervention recommendations will thus be appropriate for the aims of our study. A summary table of the synthesis approach is included in the Appendix A.

## 3. Conclusions

The rapidly changing urban environments in African countries present opportunities to protect and promote health. Public space initiatives can serve as city-scale high-impact interventions, creating rapid health gains in African cities by improving access to health-protective factors. Currently, an overview of the nature of infrastructural provisions in urban public spaces in Africa and their impact on noncommunicable disease prevention is lacking. It is necessary to understand the landscape, components, and impacts of initiatives to address health through public space initiatives in Africa. The knowledge yielded from this study will contribute to informing efforts to design public spaces for improved health in African cities.

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
