# Peer review of "A Systematic Review Protocol of Opportunities for Noncommunicable Disease Prevention via Public Space Initiatives in African Cities"

_ijerph, 2022, doi:10.3390/ijerph19042285_

Round 1

Reviewer 1 Report

This is a good protocol for a literature review. I have some specific comments:

Line 154: Are you going to get support from a librarian in structuring the search? 
Lines 147-154: It would be interesting to present the list of keywords here or in an appendix document.
Lines 155-162: There is a bit of a lack of detail in the search for grey literature, you are only going to use Google, the results may be vast. See this article to clarify this section: https://systematicreviewsjournal.biomedcentral.com/articles/10.1186/s13643-015-0125-0 

Reviewer 2 Report

This topic is based on the contradiction between rapid urbanization and health problems in Africa,which is of great value for comprehensive understanding on how public space could be used to promote leisure physical activity and, furthermore, public health. This research protocol develops considerable feasible and rational research design and detailed methods, including strategies and methods in data collection and article screening, as well as alternative analytical strategies for the expected results. Specifically, it can be improved in the following aspects:

  1. The "public space activities" in the title is inconsistent with the "leisure physical activities" in the content. The key concept of this study needs further clarification.
  2. The authors claim that adopting "mix methods" is one of its innovations. But all methods used in the article have been typical ones in literature review.
  1. The conclusion has not identified very meaningful research directions and gaps in this field. It needs to be better summarized and clarified.

Reviewer 3 Report

The paper submitted for review presents an interesting proposal for literature-based research. In my opinion, they may prove useful/value for science and for health promotion. However, it would be worthwhile to describe the significance/usefulness of the proposed research in more detail. Besides, it is necessary in my opinion to link the research procedure more clearly to the specific aims presented on page 2 (lines 82-95). To make the proposed research procedure more comprehensible, perhaps it should also be presented in a figure/chart. Since the research is to be carried out for the member countries of the African Union, I also suggest including a comparative analysis across countries/regions to recognise, among other things, the diversity of approaches to public spaces of African cities. In addition, the numbering of references (lines 260-288) should be improved and the relationship between public spaces and quality of life (wellbeing) emphasised. I am curious about the results of the study and wish you good luck with your research.
